# Contact Characteristics and Tribological Properties of the Weaving Surface of Mn-Cu and Fe-Zn Damping Alloys

**DOI:** 10.3390/ma15093303

**Published:** 2022-05-05

**Authors:** Lin Zhang, Xindong Yan, Ying Shu, Hongjuan Yang, Xiaomin Kang, Zhenbing Cai, Minhao Zhu

**Affiliations:** 1School of Mechanical and Electrical Engineering, Chengdu University of Technology, Chengdu 610059, China; hpuxindong@163.com (X.Y.); shuying20000105@163.com (Y.S.); yhj1116@126.com (H.Y.); 2School of Mechanical Engineering, University of South China, Hengyang 421001, China; kxmswjtu@163.com; 3Institute of Tribology, Southwest Jiaotong University, Chengdu 610031, China; czb_jiaoda@126.com (Z.C.); zhuminhao@139.com (M.Z.)

**Keywords:** damping alloys, tribology, contact characteristics, wear mechanism

## Abstract

In this paper, laser texturing is performed on the surface of Mn-Cu and Fe-Zn damping alloys and the tribological properties of the samples with various surface weaves under dry-sliding conditions are investigated. The results show that the surface weave parameters affect the size of the contact surface and change the number of micro-convex bodies at the contact interface. This leads to changes in the tangential damping of the contact and further affects the magnitude of the friction coefficient. Additionally, the damping properties significantly affect the wear mechanism and make it more prone to adhesive wear.

## 1. Introduction

With the development of high-speed, high-efficiency and automated mechanical equipment, the influence of vibration and noise are becoming increasingly prominent. Damping alloys receive extensive attention and research due to their good mechanical properties, which are employed in gearing under vibration and high-friction conditions [1,2,3,4]. The friction reduction effects of damping alloys lie in the weakening of vibration during the friction process [5]. The damping contact shows different tribological behaviors compared with the rigid contacts, which possess a higher friction coefficient due to the hysteresis of damping deformation [6,7,8]. The current research on the influences of damping properties during friction mainly focuses on viscoelastic materials [9,10,11] or the damping factors in interfacial contacts [12,13,14,15]. Zhang et al. [16,17,18] studied the effects of contact damping on the friction process from the perspective of fractal theory, and theoretically established the correlation between contact damping and friction process. Liu et al. [19,20] investigated the friction characteristics in the presence of damping alloy during the wear process. However, there is little research on the effects of the damping alloy as a counter-abrasive on the friction process, especially after a change in material damping.

The damping mechanism of the damping alloys is quite different from that of the viscoelastic materials [21], and the damping properties of the alloys are derived from the internal defects of the metal. The heat-treatment changes the microstructure of the damping alloys, resulting in changes in the surface hardness and Young’s modulus of the material at the same time. Therefore, it is not ideal to improve the damping properties by changing the alloy composition or using a heat-treatment process. Manganese copper alloy is a widely used damping alloy with high-damping performance and good mechanical properties, in which the martensite phase transformation of the phase twins is the source of the damping performance [22,23]. Iron-zinc alloy is another widely used damping alloy, and the source of damping in the Fe-Zn alloys is internal defects, such as dislocations [24,25]. In this study, the Mn-Cu and Fe-Zn damping alloys are selected to study the effects of the damping properties on the frictional wear process between the pairs of grinding subsets by machining the surface weave.

Previous studies showed that the surface weave affected the contact properties of the bonding surface. Arghir [26] analyzed the contact stiffness and damping of the seal ring with weaves through simulation, which showed that the weaves affected the contact stiffness and damping of the seal ring. Zahrul [27] investigated the effects of the contact density of the micro-convexity on the tangential contact damping by machining pyramidal micro-convexity weaves on the disc member, and the higher contact density led to a higher tangential contact stiffness and damping. The research of Medina [28] showed that the contact density of the micro-convexes could be controlled by changing the distance between the micro-convexes to achieve control over contact damping. Zhang [29,30,31] investigated the weaves on the contact interface and found that the surface weave affected the contact stiffness and damping between the bonding surfaces, which was explained by fractal theory.

In this work, which aim to study the effects of the damping changes on the friction process during sliding friction, the contact damping properties of the Mn-Cu and Fe-Zn damping alloys are enhanced by the micro-convexity weave on the surface. The article firstly characterizes the tribological properties of Mn-Cu, Fe-Zn damping alloys with different weave parameters and analyzes the variation of their friction coefficients, and then investigates the effect of weave parameters on the tangential damping and tangential stiffness of the friction surfaces through simulations, indicating that the variation of tangential damping affects the variation of friction coefficients. Then, the wear mechanism of the damping alloy is studied from the perspective of friction morphology, and it is found that the formation of abrasive chips has a greater influence on the wear mechanism. The effect of the change of normal force on the formation of abrasive chips of damped alloy is investigated.

## 2. Experimental Section

### 2.1. Materials

The damping alloys of Mn-Cu (Mn-20Cu-5Ni-2Fe, M5052) and Fe-Zn (27Zn-Fe, ZX09) were purchased from Shanghai Tongxiang Co., Ltd. (Shanghai, China). Table 1 shows the mechanical properties of the two alloys, and the tensile curves of the two alloys are listed in Appendix A. GCr15 steel balls with a diameter of 9.5 mm were used for the grinding sub.

### 2.2. Surface Weaving Treatment

According to previous studies, the surface weave parameters, center distance (C) and weave width (W), had a greater impact on the contact interface damping [32,33,34,35]. Weave processing by laser, and processing depth (D) was maintained to 0.1 μm (Shown in Figure 1). The weave parameters are listed in Table 2. There are 12 samples of Mn-Cu (Cu) and Fe-Zn (Fe) respectively.

### 2.3. Experimental Setup

Friction characteristics of sliding phenomena were studied in this work by performing tribological tests on CETR-UMT3 in a ball-on-block configuration. Its schematic can be seen in Figure 1. The fixture system was used to hold the specimens; the pad was held in the upper holder, which was connected with the connection part by the screw thread. A strong glue was used to fix the disc onto the metal base which would slide with the rotational motion device. During these tests, the moving stage which was controlled by the computer would move downwards and push the ball into contact with the block under a given normal load. The block would slide with the rotational motion device which was also controlled by the computer. In the test, the 2-D force sensor (sensitivity: 0.025 N; measuring range: 5–500 N) was installed to record the normal and friction forces during the whole tests. To protect the force sensor, the suspension was fixed between the connection part and the force sensor. The signals of the forces were recorded by the computer.

Before conducting the experiments, a series of preliminary tests were carried out with different values of normal loads and disc speeds. Experiments were carried out at normal force of 2 N, 5 N and 10 N (1 Hz). The test time was set to 1000 cycles for each sample.

Considering the randomness of the friction-induced vibration, each type of disc specimen was tested three times in this study to ensure the reliability of experiment results. In order to characterize interface, contact and wear behaviors of the contact interface, optical microscopy (OM) and a white light interferometer were used to examine the topographies of the disc surfaces; next, a scanning electron microscope (SEM) was used to examine the topographies of pad surfaces; finally, energy dispersive X-ray spectroscopy (EDX) analysis of the wear debris on the pads of the friction pairs was conducted after the stick-slip oscillation tests. All experiments were conducted in a controlled ambient environment (temperature of 27–30 °C and relative humidity of 60 ± 10%).

## 3. Results and Discussion

### 3.1. Friction Coefficient

The evaluation of the friction coefficient curves of the Mn-Cu and Fe-Zn samples with different center distance weavings is shown in Figure 2: both materials show two different stages of friction coefficient. The friction coefficient of the Mn-Cu samples experiences a sharp increase at the initial stage of friction and then tends to stability. The friction coefficient of the samples with a larger center distance fluctuates slightly, and the samples with a larger center distance in the final stabilization stage have smaller friction coefficients. The friction coefficient of the Fe-Zn samples decreases at the beginning of the friction process and then reaches a stable stage. Due to the higher surface hardness of Fe-Zn alloys, the fluctuation of the friction coefficient is lower than that in the Mn-Cu samples. The samples with a smaller center distance show a smaller friction coefficient, and the friction coefficients of all samples are listed in Appendix A.

Figure 3 shows the effect of the sample thickness and fabric center distance on the average friction coefficient of the sample. (The effect of sample thickness and fabric center distance on the average friction coefficient of the sample is investigated in Figure 3) The friction coefficient of the Mn-Cu samples increases with the increase in the center distance when the sample thickness is constant. The change of friction coefficient is influenced by both the sample thickness and weave center distance when the sample thickness increases. The friction coefficient of the Fe-Zn samples decreases with the increase in the sample thickness, and the friction coefficient increases as the center distance becomes larger when the thickness is constant. When the sample thickness increases, the change of center distance has a greater effect on the friction coefficient. This difference may originate from the variation of the contact state of the friction pair, the change of the weave parameters affecting the contact area, the contact strength, and the contact damping under dynamic tangential forces.

Apart from the effect of thickness, Figure 4 shows that the average friction coefficient is influenced by the weave width. The average friction coefficient increases with the increase in the weave width when the thickness of all samples is identical. There is a difference in the friction coefficients of the Mn-Cu samples and Fe-Zn samples. The average friction coefficient of the Fe-Zn and Mn-Cu samples and the sample with the narrowest weave width are about 0.15, 0.3 and 0.46, respectively.

### 3.2. Simulation of Dynamic Behavior of Surface Weaving Parameters

To investigate the effect of contact characteristics on the coefficient of friction, according to the theory of micro-convex deformation, the micro-convex body underwent three stages during the friction process, namely plastic deformation, elasto-plastic deformation, and plastic deformation [36,37,38]. When the dry friction exists between the contact surfaces, the normal stress at the contact ring boundary is small, and the shear stress tends to infinity (Figure 5). Tangential loads of any size cause the boundary of the contact zone to slide. When the tangential load is less than the maximum static friction, the contact area of a single contact point is divided into adhesion zone and slip zone. The shear deformation has a complete slip under the friction conditions mentioned in this work, but the numerical simulation results are inaccurate when the deformation is large. Therefore, only the force conditions between test and before complete slip are considered to conduct the subsequent finite element simulations [39,40].

Assumed the shear stress distribution in the adhesion zone is:(1)$$\tau_{adhesion}\left(r^{\prime }\right)=-\mu p_{0}\frac{c}{r}{\left(1-{\left(\frac{r^{\prime }}{c}\right)}^{2}\right)}^{0.5}$$

The shear stress distribution in the slip zone is:(2)$$\tau_{slip}\left(r^{\prime }\right)=\mu p_{0}{\left(1-{\left(\frac{r^{\prime }}{r}\right)}^{2}\right)}^{0.5}$$

Total tangential force of contact point is:(3)$$N_{t}=\frac{2}{3}\pi \left(\tau_{2}r^{2}-\tau_{1}c^{2}\right)=\frac{2\pi r^{2}}{3}\mu p_{0}\left(1-{\left(\frac{c}{r}\right)}^{3}\right)$$
(4)$$\frac{c}{r}={\left(1-\frac{N_{t}}{\mu N}\right)}^{\frac{1}{3}}$$

The displacement of the adhesion region is:$$\delta =\frac{1}{4E_{s}}r\pi \mu p_{0}\left(1-{\left(\frac{c}{r}\right)}^{2}\right)$$
where $\frac{1}{E_{s}}=\frac{2-\nu_{1}}{4E_{s1}}+\frac{2-\nu_{2}}{4E_{s2}}$ is the equivalent shear modulus. The contact point shear stiffness is obtained by differentiating δ.
(5)$$k_{t}^{\prime }=\frac{8E_{s}r\left(1-{\left(\frac{c}{r}\right)}^{3}\right)}{3\left(1-{\left(\frac{c}{r}\right)}^{2}\right)}$$

Macroscopically, the tangential contact stiffness of the microweave bond surface can be viewed as the sum of the contact stiffness of all micro-convexes on the weave-free region. Assuming that the number of micro-convex bodies on the microweave bonding surface is *A_n_* within the contact area, the bonding surface, due to its microweave morphology, keeps the micro-convex bodies in the weave region in an uncontacted state. Numerous studies have shown that the height of micro-convex bodies on machined surfaces obeys a Gaussian distribution [39]. Therefore, for a given surface distance d, the expected value of the number of micro-convex bodies that receive deformation due to load action in the nominal contact area is:(6)$$n=\left(1-\zeta \right)\eta A_{n}\int_{d}^{\infty }\phi \left(z\right)dz$$

ζ denotes the weave density of the microweave plane,$\;\zeta =\frac{d}{w}$ × 100% (shown in Appendix A). η denotes the density of micro-convex bodies per unit area. ϕ(z) denotes the probability density function with a normal distribution of the height of the micro-convex body on the front surface.

The microscopic morphology of the weave-free region is assumed to be isotropic, and then the interactions between the micro-convex bodies on the front surface are neglected.

According to the previous model for the calculation of contact stiffness of a single micro-convex body, the normal contact stiffness of the entire weave bond surface can be obtained as follows:(7)$$K_{n}\left(d\right)=\left(1-\zeta \right)\eta A_{n}\int_{0}^{\delta_{slip}}k_{t}^{\prime }\phi \left(z\right)dz$$

According to the research results of Mindlin [40], the energy dissipated by a single contact point in one loading cycle can be obtained as:(8)$$e_{d}^{\prime }=\frac{24{\left(\mu N\right)}^{2}}{5r}\frac{3}{4E_{s}}\Lambda ,\;\Lambda =1-{\left(1-\frac{N_{t}}{\mu N}\right)}^{\frac{5}{3}}-\frac{5N_{t}}{6\mu N}\left(1+{\left(1-\frac{N_{t}}{\mu N}\right)}^{\frac{2}{3}}\right)$$

The dissipated energy in one cycle is:(9)$$E_{d}=\left(1-\zeta \right)\eta A_{n}\int_{0}^{\delta_{slip}}e_{d}^{\prime }\phi \left(z\right)dz$$

The relevant literature shows that the dry friction damping equivalent viscous damping coefficient expression is:(10)$$C=\frac{E_{d}}{\pi \omega \delta }=\frac{4E_{s}\left(1-\zeta \right)\eta A_{n}}{\omega r\pi^{2}\mu p_{0}\left(1-{\left(\frac{c}{r}\right)}^{2}\right)}\int_{0}^{\delta_{slip}}e_{d}^{\prime }\phi \left(z\right)dz$$

$e_{d}^{\prime }$ is a function related to the friction coefficient μ. According to the literature [41,42], it can be concluded that μ~C.

The relationship between the friction coefficient and the tangential damping during the friction process is obtained by finite element modeling (Figure 6, The normal force is 2 N, the sample thickness is 0.3 μm, and the weave width is 0.1 μm). The tangential force of the Mn-Cu sample is higher, while the contact mode of the Fe-Zn sample is mainly pointing contact and its tangential force is lower, resulting in the lower friction force and friction coefficient of the Fe-Zn samples.

In order to visualize the relationship between the tangential contact damping, the average friction coefficient, and the weave structure of the samples in the simulation results, the results are shown in Figure 7. The tangential contact damping increases with the increase in the weave center distance (Figure 7c), and the friction coefficient also increases (Figure 7c). The tangential damping and friction coefficient increase when the width of the weave and the thickness of the sample increase, which exhibited the same trend as the results in the literature [41,42]. According to the Equations (3)–(10), the tangential damping is positively correlated with the actual contact surface. When the center distance of the weave increases, the actual contact area increases, leading to the increase in the tangential damping and friction coefficient. On the other hand, the sample thickness is related to the contact stiffness. When the contact area is the same and the sample thickness increases, the sample stiffness increases, resulting in the decrease in the tangential damping and friction coefficient.

In contrast to the experimental results in Figure 3 and Figure 4, the changing trend of friction coefficient change in the Fe-Zn material is in good agreement with the theoretical calculation, while only the result of weave width in the Mn-Cu material is in a better agreement with the theoretical calculation, which is related to the difference in friction process caused by the difference of the material properties.

### 3.3. Morphologies and Wear Mechanism

In the previous section, we analyzed how contact characteristics influence the friction characteristics of damping alloys. This section will show the wear morphology of the alloy. Figure 8 shows the morphologies of the Mn-Cu alloy with different weaving parameters in which the normal force is 2 N, the sample thickness is 0.3 μm, and the weave width is 0.1 μm. Figure 8a shows the sample with a center distance of 0.3 μm, Figure 8b shows the sample with a center distance of 0.4 μm, and Figure 8c shows the sample with a center distance of 0.5 μm. The friction track and wear debris are visible on the friction surface. The wear debris has two main forms, namely the smaller ones as shown in Figure 8(b1) and the larger ones in Figure 8(a1), which indicate the existence of both abrasive wear and adhesive wear mechanisms in the friction process. The adhesive wear mechanism gradually dominates with the increase in the weave center distance. Combining with the changing trend of friction coefficient, the abrasive chip accumulation has a certain influence on the change of the friction coefficient. The average friction coefficient is higher when the friction is mainly abrasive wear, while it is lower when the friction process is mainly adhesive wear, which may also be the main reason why the variation of the friction coefficient with the center distance of the weave in the experiment is not consistent with the simulation calculation.

Fe-Zn alloy exhibit different wear damage characteristics due to different material properties. Figure 9 shows the friction morphology of Fe-Zn alloy with the change of weaving. Figure 9a shows the sample with a center distance of 0.3 μm, Figure 9b shows the sample with a center distance of 0.4 μm, and Figure 9c shows the sample with a center distance of 0.5 μm. The wear surface of the sample is largely abrasive wear, and the surface is essentially a friction track parallel to the frictional direction. There is a small amount of abrasive debris in the weaving grooves. The wear mechanism is still dominated by abrasive wear with the increase in the weaving center distance. The variation of the friction coefficient of the Fe-Zn alloy is more regular and more in line with the theoretical calculation results in the absence of abrasive debris on the friction surface (Figure 9(a2,b2,c2)), which further proved that the abrasive debris is an important factor in the variation of the friction coefficient.

The morphologies and atomic distribution of the grinding balls are also characterized to further study the friction mechanism of the two materials, as shown in Figure 10. There is more material transfer on the surface of the Mn-Cu material, which indicates that the adhesive friction is more obvious in the Mn-Cu material. Furthermore, there is almost no abrasive debris accumulation and Zn elements on the grinding ball surface of the Fe-Zn material, indicating that the friction mechanism of the Fe-Zn material is mainly abrasive friction.

The abrasive chips are further investigated, focusing mainly on the accumulation of abrasive chips in the woven grooves. The Mn-Cu material has more abrasive chip accumulation, while the Fe-Zn material has less abrasive chip accumulation in the grooves, as shown in Figure 11. The main component of the abrasive chips is CuO by the analysis of EDS.

The XPS results are shown in Figure 12, and the main components of the grinding chips of both Mn-Cu and Fe-Zn materials are oxide species. The main components of the grinding chips in the Mn-Cu material are CuO and Cu_2_O (Figure 12c), which reduce the friction coefficient due to the lubricating properties of CuO [43], thus explaining the aforementioned experimental phenomena. The main component of the grinding chips in the Fe-Zn material is FeO, which has less influence on the friction interface due to the low content of wear debris and accumulation in the braided grooves.

Figure 13 shows the abrasion mark morphologies and wear cross-section of the Mn-Cu and Fe-Zn samples under a normal force of 2 N. The wear depth and volume of the Mn-Cu sample decreased with the increase in the weave center distance, and the cross-section shows a U-shape. The height of the edge of the abrasion marks is higher than that of the surface of the Mn-Cu sample, which indicates the accumulation of abrasive chips. The surface wear of the Fe-Zn sample is slight, and there is a small accumulation of abrasive chips at the edge of the abrasion marks and a more obvious scratch at the center of the abrasion marks (Figure 13(b1,b2)), which is a typical feature of abrasive wear.

The normal force (Figure 14 and Figure 15) and wear direction (Figure 16) are changed to characterize the wear properties of the samples, aiming to investigate the influence of other factors on the frictional wear of the woven samples. There is both a friction track and wear debris on the wear surface, indicating that the friction mechanism is still dominated by the abrasive wear and adhesive wear. The increase in the normal force increases the wear debris and depth, while the friction coefficient changes little, indicating that the friction becomes three-body friction, and the friction coefficient mainly depends on the properties of the wear debris.

Figure 15 shows the wear conditions of the Fe-Zn sample by changing the normal force. There is no change in the wear cross-section in the presence of the slight wear, which shows the microscopic morphology and friction coefficient. The wear marks exhibited friction tracks along the friction direction, which indicate that the wear is mainly abrasive wear. The friction coefficient increases with the increase in the normal force. The friction coefficient has a positive correlation with the tangential damping based on the previous analysis, and the increase in the normal force increases the tangential damping of the woven sample, thus increasing its friction coefficient.

Figure 16 shows the Mn-Cu friction samples with varying normal forces under the radial conditions. Compared with the sample perpendicular to the weaving direction, the friction coefficient of the sample is lower under the same normal force conditions, and the abrasive chips are more likely to enter the grooves and adhere to the wear surface. However, the wear amount of the Mn-Cu friction sample is not much different from that of the sample perpendicular to the grooves. Other data are included in Appendix A.

### 3.4. Discussion

The intrinsic properties and contact characteristics of the materials have a great influence on the frictional wear performance of the damping alloys based on the aforementioned data. The variation of the weave parameters doesn’t affect the surface hardness, Young’s modulus of the material, or the surface roughness of the sample, but it does change the tangential damping of the grinding pair by changing the contact area, thus affecting the friction coefficient of the sample. A larger contact area means that the number of micro-convex bodies in contact with the counter-abrasive pair at the contact interface increases, and the tangential damping becomes larger, leading to an increase in the friction coefficient, which can be predictable in the absence of abrasive chips affecting the friction process. The difference in the material damping also plays a great role in the wear process. The frictional wear performance of the Mn-Cu samples and Fe-Zn samples with large differences in the damping properties have different wear properties. The damping properties of the Mn-Cu materials originate from their abundant internal defects, which dissipate mechanical energy by converting it into thermal energy under the dynamic loading [44,45,46]. On the one hand, the Mn-Cu material with abundant internal defects has a higher damping friction coefficient under the action of friction. On the other hand, the internal defects of the Mn-Cu material make the material more prone to wear, produce abrasive chips, and adhere to the surface of the wear marks, which all affect the process of wear. The internal defects of the Fe-Zn materials are relatively less, and the damping performance is lower than that of the Mn-Cu material. During the friction process, the material generates shear failure caused by the shear force and produces hard wear debris, which is more easily discharged from the wear surface rather than adhered to the wear surface. The abrasive wear of the materials with poor damping performance (Fe-Zn alloy) is the main wear mode, as shown in Figure 17.

## 4. Conclusions

In this work, a series of damping alloy samples with different thicknesses and weave sizes are prepared by laser processing, and the tribological properties of the samples are characterized under sliding conditions. The following conclusions are obtained:(1)The damping properties, sample thickness, and weave parameters of the damping alloys are the main factors affecting the friction properties of the material. The damping performance of the Mn-Cu sample is better than that of the Fe-Zn sample, while the friction coefficient of the Mn-Cu sample is greater than that of the Fe-Zn material under the same friction parameters, and the wear damage of the Mn-Cu sample is more severe than that of the Fe-Zn sample. The influence of the weaving parameters on the tribological performance is reflected in the variability of the contact area. According to the theoretical calculation results, the material thickness affects the contact stiffness between the abrasive pairs, and the greater the contact stiffness is, the less likely the material will be deformed under the action of tangential forces and produce wear.(2)The differences in the weave parameters leads to differences in the tangential damping of the samples, and thus the difference in the coefficient of friction, which essentially results from the changes in the number of micro-convex bodies on the contact surface caused by the changes of the contact area.(3)The tangential damping theory is more applicable to alloy materials with reduced damping properties. When the friction pair is composed of alloy materials with higher damping properties, the wear mechanism is favorable to adhesive wear rather than abrasive wear, which is due to the abundant defects inside the damping alloy, resulting in the micro-convex bodies being more likely to rupture and form wear debris under the tangential forces. The wear debris adheres easily to the wear surface, leading to changes in the friction coefficient. CuO debris formed by friction in the Mn-Cu material acts as the lubrication at the friction interface, leading to the reduction of the friction coefficient.

## Figures and Tables

**Figure 1 materials-15-03303-f001:**
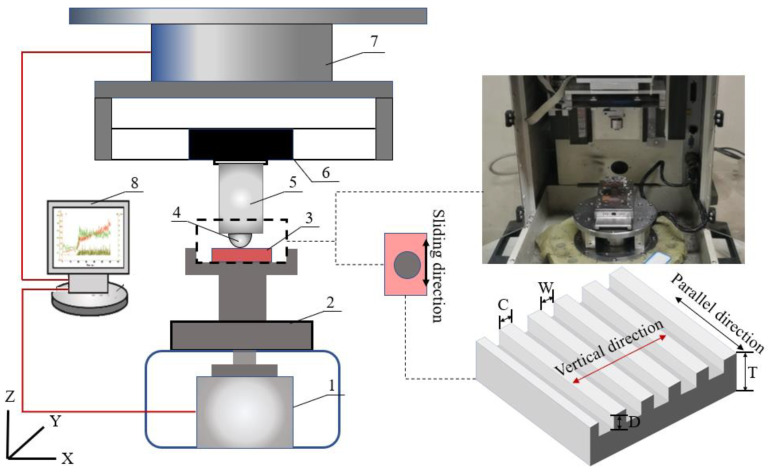
Schematic diagram of the test setup: (1) rotary motor, (2) sample table, (3) block specimen, (4) ball specimen, (5) ball holder, (6) suspension, (7) 2-D strain-gauge force sensor, (8) computer.

**Figure 2 materials-15-03303-f002:**
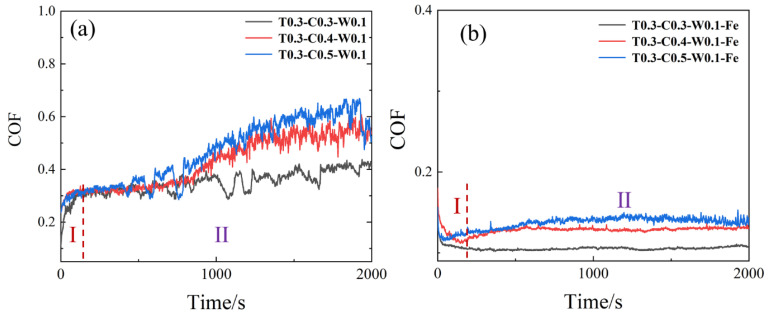
Friction coefficient curves of (**a**) Mn-Cu and (**b**) Fe-Zn samples with different center distance weavings.

**Figure 3 materials-15-03303-f003:**
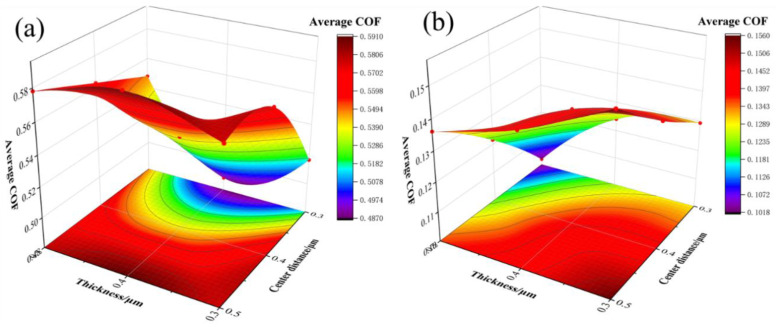
Average friction coefficient influenced by sample thickness and center distance of weave (**a**) Mn-Cu samples, (**b**) Fe-Zn samples.

**Figure 4 materials-15-03303-f004:**
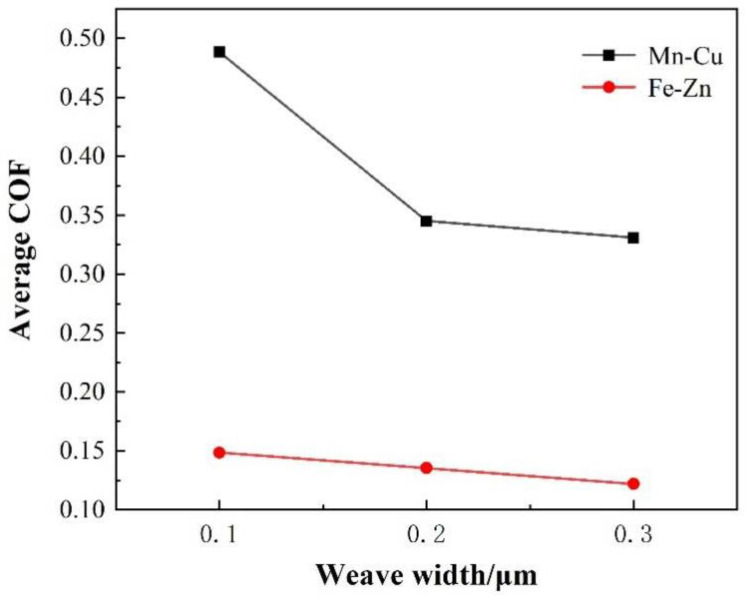
Average friction coefficients of the Mn-Cu and Fe-Zn samples with the different weave width.

**Figure 5 materials-15-03303-f005:**
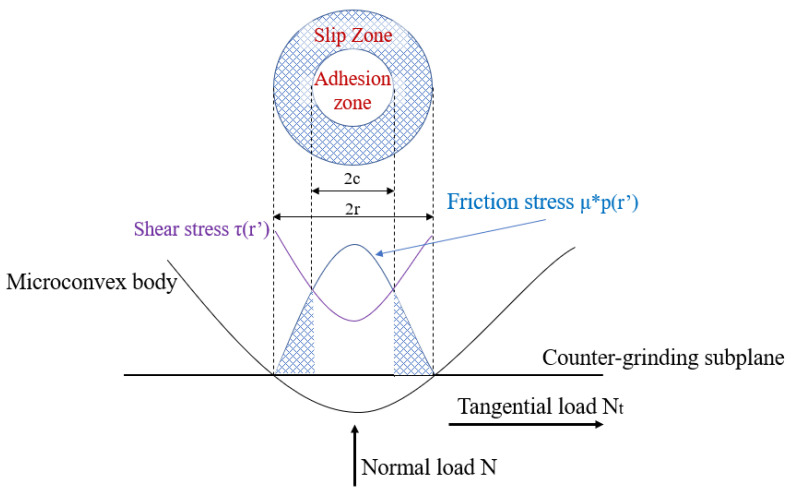
Micro-convexity’s tangential contact considering friction force.

**Figure 6 materials-15-03303-f006:**
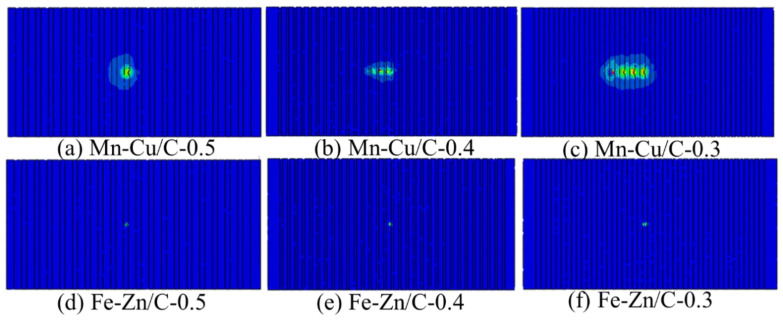
Friction process simulation of Mn-Cu and Fe-Zn with different weave center distance.

**Figure 7 materials-15-03303-f007:**
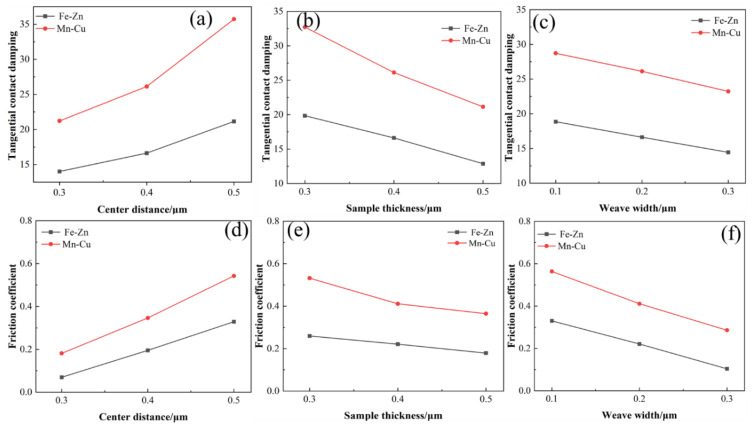
Tangential contact damping of the samples. (**a**) The relationship between Tangential contact damping and center distance. (**b**) The relationship between Tangential contact damping and Sample thickness. (**c**) The relationship between tangential contact damping and weave width. (**d**) The relationship between Friction coefficient and center distance. (**e**) The relationship between Friction coefficient and Sample thickness. (**f**) The relationship between Friction coefficient and Weave wideth.

**Figure 8 materials-15-03303-f008:**
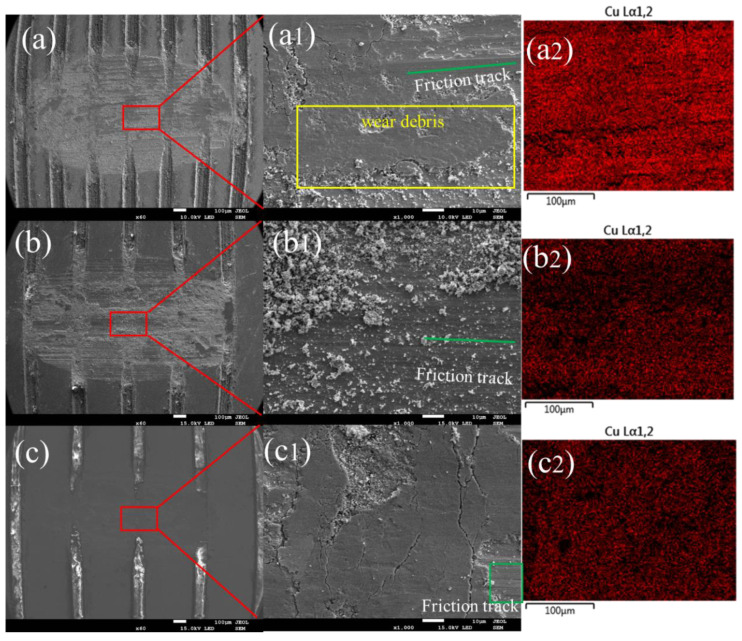
SEM images and Cu elemental mappings of the worn surfaces of Mn-Cu samples (T0.3-W0.1-N2 group). (**a**) The sample with a center distance of 0.3 μm. (**b**) Shows the sample with a center distance of 0.4 μm. (**c**) Shows the sample with a center distance of 0.5 μm. (**a1**) Partial enlargement of a-graph. (**b1**) Partial enlargement of a-graph. (**c1**) Partial enlargement of c-graph. (**a2**,**b2**,**c2**) Copper contents different friction surfaces.

**Figure 9 materials-15-03303-f009:**
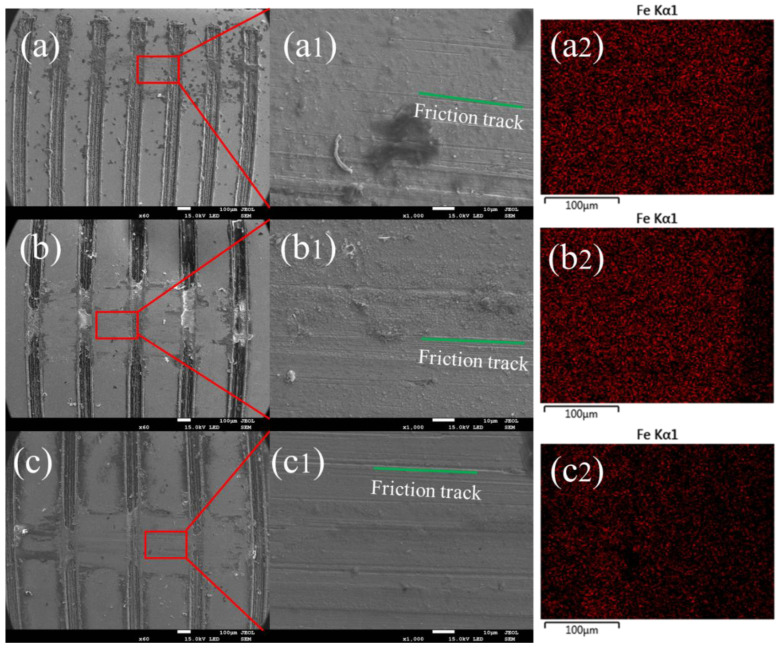
Elemental mapping of Fe on the worn surfaces of Fe-Zn samples (T0.3-W0.1-N2-Fe group). (**a**) The sample with a center distance of 0.3 μm. (**b**) The sample with a center distance of 0.4 μm. (**c**) The sample with a center distance of 0.5 μm. (**a1**) Partial enlargement of a-graph. (**b1**) Partial enlargement of a-graph. (**c1**) Partial enlargement of c-graph. (**a2**,**b2**,**c2**) Iron contents different friction surfaces.

**Figure 10 materials-15-03303-f010:**
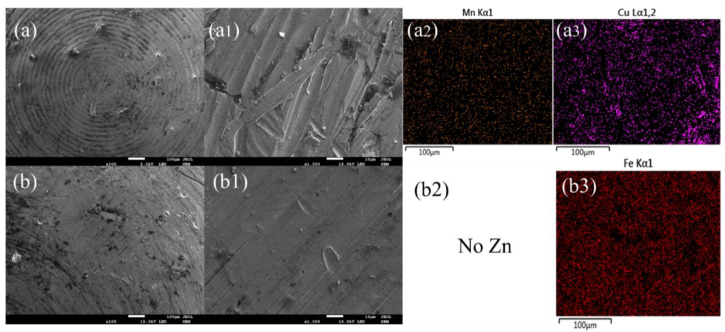
Elemental mapping of Mn, Cu, Fe, Zn on the worn surfaces of counterfaces. (**a**,**b**,**a1**,**b1**) Morphology and atomic distribution of grinding ball. (**a2**) Enlarged view of the surface of manganese containing material. (**a3**) Enlarged view of copper containing material surface. (**b2**) Enlarged view of zinc free material surface. (**b3**) Enlarged view of ferrous material surface.

**Figure 11 materials-15-03303-f011:**
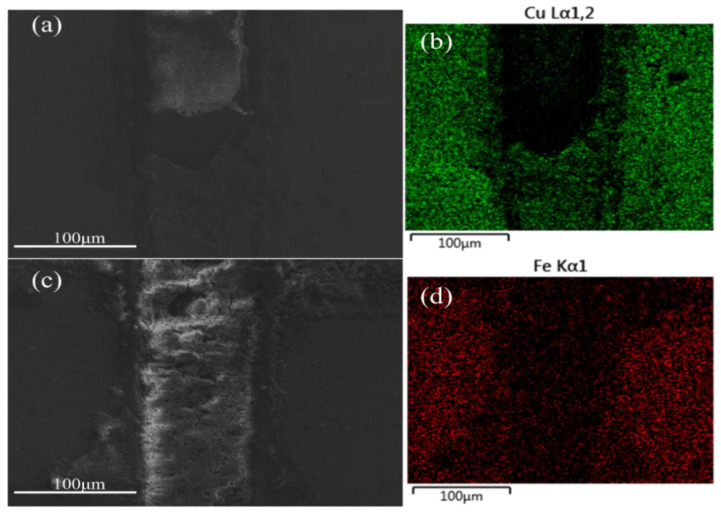
(**a**) Wear debris in grooves of Cu. (**b**) Elemental mapping of Cu. (**c**) Wear debris in grooves of Fe. (**d**) Elemental mapping of Fe.

**Figure 12 materials-15-03303-f012:**
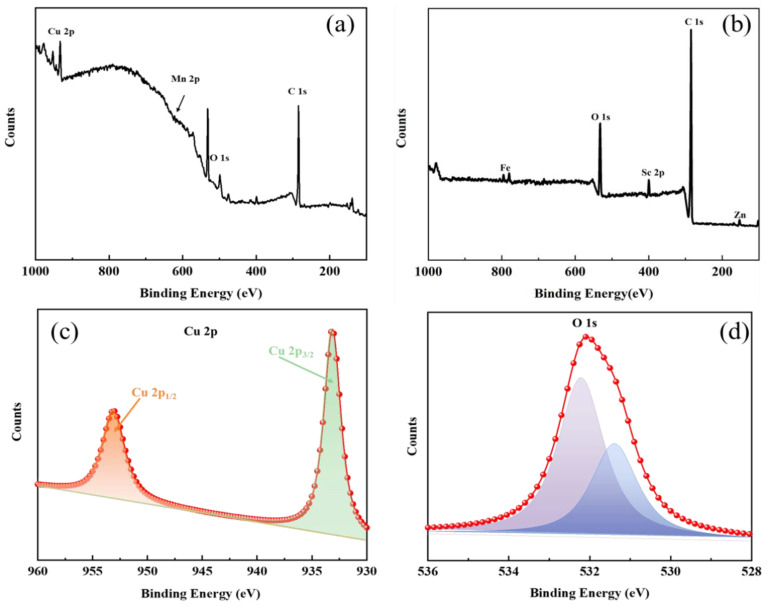
XPS spectrum of sample Mn-Cu (**a**) and Fe-Zn (**b**), Cu 2p (**c**) of Mn-Cu sample and O 1s (**d**) of Mn-Cu sample.

**Figure 13 materials-15-03303-f013:**
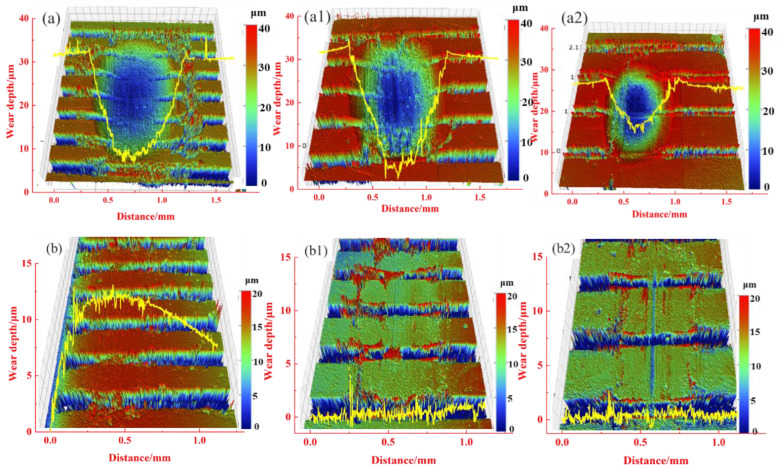
Morphology of the abrasion marks of Mn-Cu samples (**a**) T0.3-W0.1-C0.3, (**a1**) T0.3-W0.1-C0.4, (**a2**) T0.3-W0.1-C0.4 and Fe-Zn samples (**b**) T0.3-W0.1-C0.3, (**b1**) T0.3-W0.1-C0.4, (**b2**) T0.3-W0.1-C0.5.

**Figure 14 materials-15-03303-f014:**
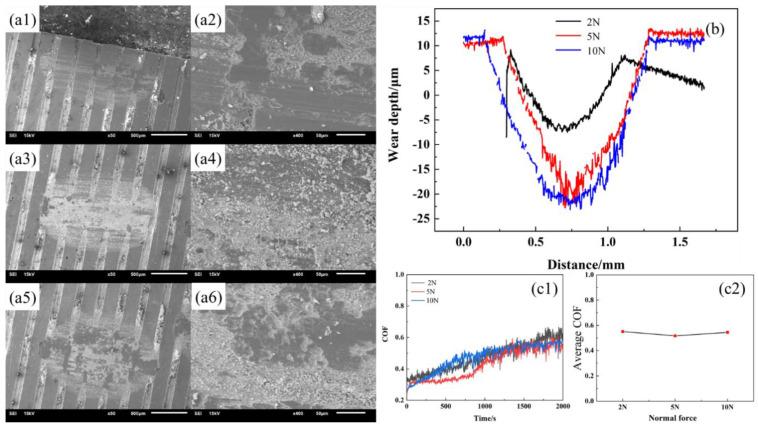
Morphology of Mn-Cu samples (**a1**) T0.3-W0.1-C0.3-2N, (**a2**) Enlarged view of a1 (**a3**) T0.3-W0.1-C0.3-5N, (**a4**) Enlarged view of a3, (**a5**) T0.3-W0.1-C0.3-10N, (**a6**) Enlarged view of (**a5**); (**b**) Wear depth of samples; (**c1**,**c2**) Coefficient of friction of samples.

**Figure 15 materials-15-03303-f015:**
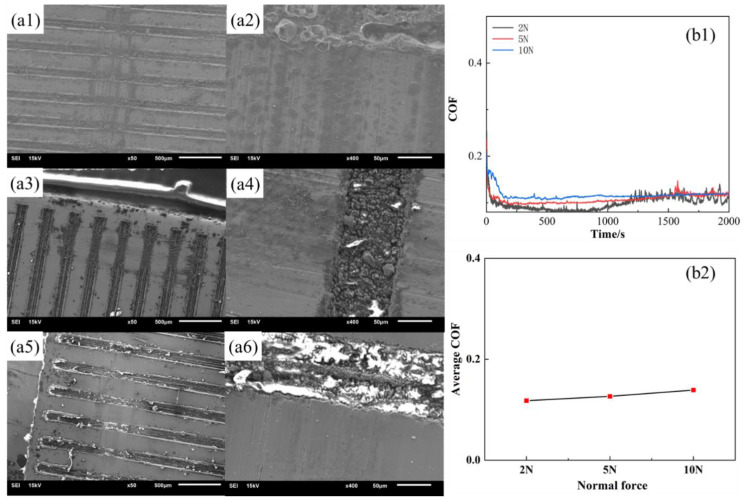
Morphology of Fe-Zn samples (**a1**) T0.3-W0.1-C0.3-2N, (**a2**) Enlarged view of a1, (**a3**) T0.3-W0.1-C0.3-5N, (**a4**) Enlarged view of a3, (**a5**) T0.3-W0.1-C0.3-10N, (**a6**) Enlarged view of a5; (**b1**,**b2**) Coefficient of friction of samples.

**Figure 16 materials-15-03303-f016:**
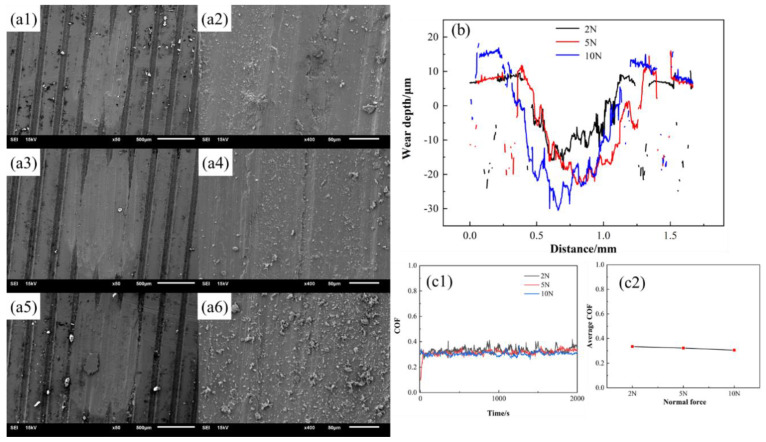
Morphology of Mn-Cu samples with parallel friction direction (**a1**) T0.3-W0.1-C0.3-2N, (**a2**) Enlarged view of a1, (**a3**) T0.3-W0.1-C0.3-5N, (**a4**) Enlarged view of a3, (**a5**) T0.3-W0.1-C0.3-10N, (**a6**) Enlarged view of a5; (**b**) Wear depth of samples; (**c1**,**c2**) Coefficient of friction of samples.

**Figure 17 materials-15-03303-f017:**
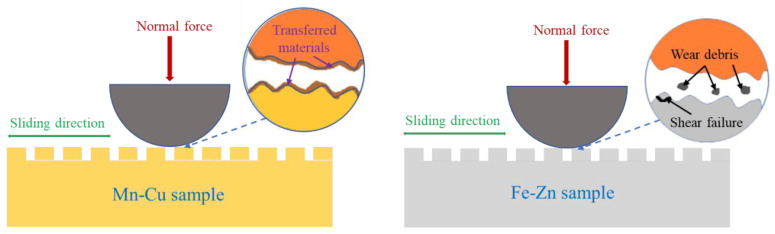
Schematic diagrams of wear mechanism under dry-sliding conditions of Mn-Cu and Fe-Zn woven samples.

**Table 1 materials-15-03303-t001:** The mechanical properties of frictional sub-materials.

	Elastic Modulus (GPa)	Tensile Strength (MPa)	Hardness (Hv)	Damping Property (Tan δ)
Mn-Cu alloy	11.5	40.4	39.2	0.56
Fe-Zn alloy	25.8	133.2	70.1	0.21
GCr15	207.0	861.0	890.0	

**Table 2 materials-15-03303-t002:** The weaving parameters of samples.

	Thickness (T)/μm	Center Distance (C)/μm	Weave Width (W)/μm
T0.3	0.3	0.3, 0.4, 0.5	0.1
T0.4	0.4	0.3, 0.4, 0.5	0.1
T0.5	0.5	0.3, 0.4, 0.5	0.1
T0.4	0.4	0.4	0.1, 0.2, 0.3

## Data Availability

No applicable.

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
