# Peer review of "Contact Characteristics and Tribological Properties of the Weaving Surface of Mn-Cu and Fe-Zn Damping Alloys"

_materials, 2022, doi:10.3390/ma15093303_

Round 1

Reviewer 1 Report

This manuscript studies contact characteristics and tribological properties of the weaving surface of Mn-Cu and Fe-Zn damping alloys. The research purpose is interesting and it would be interesting to the potential readers of Materials. However, the manuscript has many critical flaws. 

1) In the introduction, some sentences are not clear. Please, rewrite and reorganize. Especially, the paragraph explaining previous research is not connected to other paragraphs. It is difficult to understand how those previous articles (from 26 to 31) provide inspiration or give research backgrounds. 

2) It is curious the unit of length in Table 2 (here, micronmeter) is right? The scale bar on SEM images looks the unit of length should be 'mm'

3) There are many misspells, such as "plaies". Furthermore, there are a lot of spacing mistakes. 

4) Please, provide FEM simulation conditions in detail.

5)  What are the initial microstructures of those alloys? 

6)  The length of the manuscript is adequate, but there are many figures. The last figure and its explanation would be not necessary for this article. 

7) Scale bars and their units in SEM images are not clear. Please, organize them. 

8) The equations for the analytical solution are not important for this manuscript. It would be better to focus on FEM anlaysis. 

9) The deformed area in Figure 6 should be enlarged, and the authors should state which field variable is shown in Figure 6. 

10) Figure 7 is obtained by FEM, so, the authors should clarify it in the figure caption.

11) The region of wear debris seems to be an oxidized layer, and the oxidized layer looks randomly distributed in Figures 8a, 8b, and 8c.  As the authors mentioned, the detached chips could be accumulated on the weave surface as well. It would be better to give depth discussions on the formation of worn surfaces during the sliding wear. 

Author Response

This manuscript studies contact characteristics and tribological properties of the weaving surface of Mn-Cu and Fe-Zn damping alloys. The research purpose is interesting and it would be interesting to the potential readers of Materials. However, the manuscript has many critical flaws.

  • In the introduction, some sentences are not clear. Please, rewrite and reorganize. Especially, the paragraph explaining previous research is not connected to other paragraphs. It is difficult to understand how those previous articles (from 26 to 31) provide inspiration or give research backgrounds.

We reorganized the sentences in Introduction and provided the clarification of the unclear points, because it is very difficult to study the effect of changes in the damping characteristics of the material on the sliding friction of the material. Methods such as heat treatment change the microstructure of the alloy material so that the surface hardness, Young's modulus of the alloy changes at the same time. So, it is necessary to find a way to change the contact damping of the alloy while maintaining its properties.

  • It is curious the unit of length in Table 2 (here, micronmeter) is right? The scale bar on SEM images looks the unit of length should be 'mm'

The unit here is indeed mm, thanks for pointing out the error, the relevant expression in the text has also been corrected

  • There are many misspells, such as "plaies". Furthermore, there are a lot of spacing mistakes.

The relevant statements in the text have also been corrected

  • Please, provide FEM simulation conditions in detail.

Flat plate material sample set as solid, set material mechanical property parameters of material sample (Fe-Zn, Mn-Cu), mainly including density, elastic modulus, Poisson's ratio, stress-strain material parameters, etc.; small ball set as rigid body.

Then input boundary conditions: the bottom end of the plate is set with 6 degrees of freedom full constraints, the reference point of the sphere is set as force load, and the contact between the sphere and the plate is set.

  • What are the initial microstructures of those alloys?

The damping alloys of Mn-Cu (Mn-20Cu-5Ni-2Fe, M5052) and Fe-Zn (27Zn-Fe, ZX09) were purchased from Shanghai Tongxiang Co. Ltd. Mn-Cu alloy undergoes martensitic phase transformation, producing a large number of twin and sub-twin structures; In contrast, Fe-Zn alloys mainly use dislocations at grain boundaries as the source of damping properties. In this paper, we focus on the macroscopic differences in the damping characteristics and surface hardness of the two alloys due to the different damping mechanisms.

  • The length of the manuscript is adequate, but there are many figures. The last figure and its explanation would be not necessary for this article.

The last figure has been deleted.

  • Scale bars and their units in SEM images are not clear. Please, organize them.

The picture in the text has been replaced.

  • The equations for the analytical solution are not important for this manuscript. It would be better to focus on FEM anlaysis.

Some formula derivation procedures have been removed. Added some discussion on the results of the finite element simulation.

In contrast to the experimental results in Fig. 3. and Fig. 4., the changing trend of friction coefficient change in the Fe-Zn material is in good agreement with the theoretical calculation, while only the result of weave width in the Mn-Cu material is in a better agreement with the theoretical calculation, which is related to the difference in friction process caused by the difference of the material properties.

According to the results of finite element simulation, the deformation range and deformation of Mn-Cu material under the same force are relatively larger, which is caused by the higher Tanδ and lower contact stiffness of Mn-Cu material. The wear surface of Mn-Cu material will also be larger in the actual friction process, which may also be one of the main reasons for the large fluctuation of friction coefficient of Mn-Cu material.

  • The deformed area in Figure 6 should be enlarged, and the authors should state which field variable is shown in Figure 6.

The deformed area in Figure 6 has be enlarged, and figure 6 shows the stress field.

  • Figure 7 is obtained by FEM, so, the authors should clarify it in the figure caption.

The picture has been relabeled.

  • The region of wear debris seems to be an oxidized layer, and the oxidized layer looks randomly distributed in Figures 8a, 8b, and 8c. As the authors mentioned, the detached chips could be accumulated on the weave surface as well. It would be better to give depth discussions on the formation of worn surfaces during the sliding wear.

The formation of sliding wear surfaces is described in the DISCUSS section, and in the revised version we describe the surface formation mechanism in more detail. The Mn-Cu material with abundant internal defects have a higher damping friction coefficient under the action of friction. On the other hand, the internal defects of the Mn-Cu material make the material more prone to wear, produce abrasive chips, and adhere to the surface of the wear marks, which affect the process of wear. The internal defects of the Fe-Zn materials are relatively less, and the damping performance is lower than that of the Mn-Cu material. During the friction process, the material generates shear failure caused by the shear force and produces hard wear debris, which is more easily discharged from the wear surface rather than adhered to the wear surface.

Reviewer 2 Report

the manuscript is presenting an interesting topic on developing the quality of surface texting using a laser. 

A few questions and comments are due before we proceed with this submission:

  1. in the sample preparation section. you decided the test time to be set in 1000 cycles for each sample. how did you come to this number?
  2. Why do authors believe that the higher surface hardness results in less fluctuation of the friction coefficient? please mention the relation of the two-parameter by the equation.
  3. the laser application must be cited to https://doi.org/10.1016/j.rinp.2019.102883
  4. explain in relation to fig. 4 why the boundary of the contact zone slides?
  5.  the variation of the friction coefficient with the center distance of the weave in the experiment is not consistent with the simulation
    calculation. please explain in relation to the formulation and fig. 3 and 4 and correlate them with fig. 8.

when authors take my comments in , i can reconsider my decision.

Author Response

  1. In the sample preparation section. you decided the test time to be set in 1000 cycles for each sample. how did you come to this number?

We conducted a series of pre-experiments in conducting the study, testing the samples for 100-5000 cycles, and we have chosen to discuss in depth the results of only 1000 tests where the test phenomenon is more prominent.

  1. Why do authors believe that the higher surface hardness results in less fluctuation of the friction coefficient? please mention the relation of the two-parameter by the equation.

We draw the relevant conclusions from the experimental phenomena. From the results of the friction coefficient, the fluctuation of the friction coefficient of the Fe-Zn sample is much smaller than that of the Mn-Cu sample. We believe that this is due to the surface hardness of the two materials, due to the higher hardness of Fe-Zn, there will not be a large local deformation, so the friction coefficient is more stable.

  1. the laser application must be cited to https://doi.org/10.1016/j.rinp.2019.102883

Related literature has been cited

  1. explain in relation to fig. 4 why the boundary of the contact zone slides?

The average friction coefficient of the samples decreases with the increase of the center distance of the sample weaving, which is due to the decrease of the contact surface and the increase of the normal contact stress, and the edge contact region of the microconvex body is more difficult to slip, while for the Mn-Cu material, which has less hardness and high contact damping, the edge contact region is more likely to slip under the same contact stress conditions, resulting in a larger average friction coefficient than that of the Fe-Zn material.

5.the variation of the friction coefficient with the center distance of the weave in the experiment is not consistent with the simulation calculation. please explain in relation to the formulation and fig. 3 and 4 and correlate them with fig. 8.

The agreement between the experimental results and the trend of the simulation calculation results for Fe-Zn material is higher than that for Mn-Cu material, mainly because the internal defects of Mn-Cu material make the material more prone to wear, generate abrasive chips, and adhere to the surface of the wear marks, which affect the wear process. The Fe-Zn material, on the other hand, has relatively few internal defects and lower damping performance than the Mn-Cu material. During the friction process, the material is shear damaged by the shear force, producing hard wear debris, which are more easily discharged from the wear surface rather than adhering to the wear surface. The abrasive debris adhering to the wear surface affects the variation of the friction coefficient, leading to the inconsistency between the simulation results and the experimental results.

Round 2

Reviewer 2 Report

the authors answered some parts of my previous comments but not properly inserted the comments in the paper. few changes were applied only.

the new comments are obligatory to implement as otherwise, the paper will have to be rejected right away.

1) please provide evidence either from your experiments and data analysis on effect on friction coefficient due to sample thickness increase?

2) why the contact area is the same but the sample thickness increases. the equation for this has been indicated by requires explanation.

3) the optimum point of fig. 4 has to be explained in terms of morphology as described in 3.3. there is least correlation between these two sections.

4) the reference is required to https://doi.org/10.1016/j.rinp.2019.102883

when authors take my comments in, i can reconsider my  decision.

Author Response

  • please provide evidence either from your experiments and data analysis on effect on friction coefficient due to sample thickness increase?

The evidence on effect on friction coefficient due to sample thickness increase is shown in the paper as Fig. 3, there is the change of the friction coefficient for each test point in the supply material. In the figure we can see that the change of the average friction coefficient for the Mn-Cu sample seems to be not so regular, while for the Fe-Zn alloy, with the increase of the sample thickness, the average friction coefficient decreases, and from the analysis of the contact stiffness model, the increase of sample thickness improves the contact stiffness of the sample, which makes it more difficult for the micro-convex body to produce deformation during the friction process, and the friction process will become more stable, which is also confirmed from the actual testing process of Fe-Zn samples. As for the Mn-Cu alloy, the average friction coefficient changes due to the abrasive chips adhering to the friction surface, resulting in an irregular effect of sample thickness on the average friction coefficient.

2) why the contact area is the same but the sample thickness increases. the equation for this has been indicated by requires explanation.

I am very sorry, but I cannot understand this question, according to the contact mode of the friction pair described in Figure 1, the contact area is only related to the shape parameters of the surface weave, and according to the theory of contact stiffness covered in the literature and in the text [40-42], an increase in the thickness of the sample leads to an increase in the contact stiffness of the two pairs of grinding pairs.

3) the optimum point of fig. 4 has to be explained in terms of morphology as described in 3.3. there is least correlation between these two sections.

The tangential damping theory is more applicable to alloy materials with less damping properties. When the friction pair is composed of alloy materials with higher damping properties, the wear mechanism is favorable to adhesive wear rather than abrasive wear, which is due to the abundant defects inside the damping alloy, resulting in the micro-convex bodies being more likely to rupture and form wear debris under the tangential forces. The wear debris is easier to adhere to the wear surface, leading to the changes in the friction coefficient. CuO debris formed by friction in the Mn-Cu material act as the lubrication at the friction interface, leading to the reduction of the friction coefficient.

4) the reference is required to https://doi.org/10.1016/j.rinp.2019.102883

Related literature has been cited. In fact, in the previous modified version, the literature was included.

Round 3

Reviewer 2 Report

my comments were implemented properly, i can accept it for publication now